# Navigating the Nexus: HIV and Breast Cancer—A Critical Review

**DOI:** 10.3390/ijms25063222

**Published:** 2024-03-12

**Authors:** Andrea Marino, Giuliana Pavone, Federica Martorana, Viviana Fisicaro, Lucia Motta, Serena Spampinato, Benedetto Maurizio Celesia, Bruno Cacopardo, Paolo Vigneri, Giuseppe Nunnari

**Affiliations:** 1Unit of Infectious Diseases, Department of Clinical and Experimental Medicine, ARNAS Garibaldi Hospital, University of Catania, 95123 Catania, Italy; andrea.marino@unict.it (A.M.); bmcelesia@gmail.com (B.M.C.); cacopard@unict.it (B.C.); giuseppe.nunnari1@unict.it (G.N.); 2Medical Oncology Unit, Humanitas Istituto Clinico Catanese, 95045 Catania, Italy; luciamotta693@gmail.com (L.M.); vigneripaolo@gmail.com (P.V.); 3Department of Clinical and Experimental Medicine, University of Catania, 95123 Catania, Italy; federica.martorana@unict.it; 4Department of Clinical and Experimental Medicine, University of Messina, 98124 Messina, Italy; viviana.fisicaro@gmail.com (V.F.); serenaspampinato93@gmail.com (S.S.)

**Keywords:** HIV, breast cancer, non-AIDS defining cancers, AIDS, drug–drug interactions

## Abstract

Despite significant advances in the management of antiretroviral therapy (ART), leading to improved life expectancy for people living with HIV (PLWH), the incidence of non-AIDS-defining cancers, including breast cancer, has emerged as a critical concern. This review synthesizes current evidence on the epidemiology of breast cancer among HIV-infected individuals, highlighting the potential for an altered risk profile, earlier onset, and more advanced disease at diagnosis. It delves into the molecular considerations underpinning the relationship between HIV and breast cancer, including the role of immunosuppression, chronic inflammation, and gene expression alterations. Additionally, it examines the complexities of managing breast cancer in the context of HIV, particularly the challenges posed by ART and anticancer agents’ cross-toxicities and drug–drug interactions. The review also addresses survival disparities, underscoring the need for improved cancer care in this population. By identifying gaps in knowledge and areas requiring further research, this review aims to illuminate the complexities of HIV-associated breast cancer, fostering a deeper understanding of its epidemiology, molecular basis, and clinical management challenges, thereby contributing to better outcomes for individuals at the intersection of these two conditions. This narrative review systematically explores the intersection of HIV infection and breast cancer, focusing on the impact of HIV on breast cancer risk, outcomes, and treatment challenges.

## 1. Introduction

Breast cancer (BC) remains one of the most prevalent and deadly cancers among women globally, affecting millions and leading as the cause of cancer-related deaths among the female population, with 2.3 million new cancer cases (one in four new cancer cases) and 685,000 cancer deaths (one in six deaths) in 2020 [1]. Human Immunodeficiency Virus (HIV) remains a critical public health issue, with approximately 39 million people living with the virus at the end of 2022 and 1.3 million new diagnoses [2]. The advent of combination antiretroviral therapy (ART) has dramatically transformed the landscape of HIV infection, converting it from a fatal disease to a manageable chronic condition [3]. As the lifespan of people living with HIV (PLWH) extends, non-AIDS-defining cancers (NADCs), including BC, have emerged as critical components of their long-term health outlook. In the overall population of BC patients, regardless of HIV status, women with HIV made up less than 1% of cases. However, this percentage increased in specific regions of Africa: in Eastern, Western, and Middle Africa, they represented 4–6% of BC cases among women younger than 50 years. Moreover, in Southern Africa, this group accounted for 26% of BC diagnoses in women under 50 years old [4].

The pathophysiology of HIV involves a systematic assault on the immune system, primarily targeting CD4+ T cells, which leads to a progressive decline in immune competence [5]. Immune system compromise not only predisposes individuals to opportunistic infections but might also alter the natural history of various NADCs, including BC. Despite advances in ART, with several new drugs clinicians could use, the implications of HIV on cancer risk and cancer-related outcomes remain a critical area and a clinical challenge [6]. Emerging research suggests that individuals with HIV may be at an altered risk for developing BC, although the data remain inconclusive [7]. Several theoretical frameworks have been proposed to explain this potential link, including immune surveillance impairment, chronic inflammation, and the role of oncogenic viruses, which are more prevalent in individuals with HIV [7]. However, studies exploring the incidence of BC in HIV-infected populations have yielded mixed results, underscoring the need for a comprehensive review of the literature to understand the nuances of this relationship [8].

## 2. Epidemiological and Demographic Considerations

Assessing the epidemiology of BC in PLWH could be challenging, primarily due to the intricate interplay between the underlying immunodeficiency caused by HIV and the multifactorial nature of breast cancer. Different geographical cohorts from around the world illustrate different standard incidence ratios (SIRs) or odds ratios (ORs) for breast cancer among people living with HIV compared to the general population (Figure 1) [9]. This complexity is compounded by several key factors [4].

Firstly, this epidemiology involves the overlap and interaction of risk factors for both HIV infection and BC. Age, genetics, lifestyle choices, and exposure to other environmental risks can independently and synergistically influence the risk of developing BC [10]. In PLWH, these factors may interact differently due to different cut-offs for aging, different levels of inflammation, and different incidences of comorbidities, especially those related to cardiovascular and metabolic systems, further complicating the epidemiological analysis [4]. Moreover, as BC risk increases with age, and considering the improved PLWH lifespan thanks to ART efficacy, this demographic shift introduces new complexities in understanding the incidence and characteristics of BC in this group [11]. The availability and quality of epidemiological data also pose significant challenges. In many regions, especially where HIV is most prevalent, there may be limited cancer registries and inadequate healthcare infrastructure for comprehensive cancer screening and diagnosis. This limitation leads to underreporting and underdiagnosis, skewing epidemiological data. Furthermore, the stigma associated with HIV can lead to delayed cancer screening and diagnosis, as individuals may avoid seeking healthcare. This delay can result in a presentation at more advanced stages of breast cancer, which complicates the assessment of its true epidemiology for PLWH [12].

The interplay between HIV infection and BC has been a focal point of medical research, reflecting the complexities inherent in the intersection of infectious diseases and cancer epidemiology [13]. Initially, during the early stages of the HIV epidemic, BC incidence among PLWH was reported to be lower than in the general population. This observation was attributed to the shorter life expectancy of individuals with HIV and the dominance of aggressive AIDS-defining cancers, overshadowing the incidence of other cancers [14]. However, the landscape of HIV treatment and management has dramatically changed with the advent and widespread adoption of ART, leading to an aging population and a corresponding increase in the incidence of age-related comorbidities along with non-AIDS-defining cancers, including BC [4].

Robbins et al. [15] used Poisson models on US HIV/AIDS Cancer Match Study data, collected from 1996 to 2013, to calculate changes in breast cancer incidence rates, accounting for demographics and other trends. They found that breast cancer rates stayed stable over time, but demographic changes contributed to an increasing trend in incidence.

This heightened risk is particularly noteworthy as it underscores the impact of HIV on cancer development beyond the traditional AIDS-defining cancers. Also, Coghill et al. [16] identified that women with HIV are at a modestly increased risk of developing BC, pointing to the significant roles of immune suppression and chronic inflammation in this elevated risk. This discovery highlights the need for further research into the mechanisms by which HIV infection alters cancer risk. Conversely, the research by Shiels et al. [17] presented a contrasting perspective. It indicated that while the overall cancer risk is higher for PLWH, the incidence of BC specifically did not significantly differ from that in uninfected women. This paradox underscores the complex nature of cancer epidemiology within the context of HIV and suggests that the effects of ART and improved healthcare may be mitigating the increased cancer risk associated with HIV.

The significance of traditional BC risk factors such as hormonal influences, genetics, and lifestyle factors [18,19] and their interaction with HIV-related immunological changes cannot be overstated. The interplay between these factors is complex and requires in-depth exploration. For instance, the hormonal changes induced by HIV and its treatments may alter the traditional pathways through which BC develops, necessitating a reevaluation of BC risk assessment models in HIV-infected individuals [20]. Geographical and demographic disparities in BC incidence among PLWH further complicate this epidemiological picture. Studies have consistently shown that in regions with high HIV prevalence, such as sub-Saharan Africa, the incidence and characteristics of BC in HIV+ve patients vary significantly from those in regions with lower HIV prevalence. The study by Cubasch et al. [21] in South Africa found that HIV+ve women had a younger median age at BC diagnosis compared to their HIV-ve counterparts. This difference suggests that HIV may impact the age of onset and the biological behavior of BC. In more developed regions, where access to ART and HIV management is more robust, the incidence of BC among PLWH tends to align more closely with that of the general population. This trend was highlighted in a European cohort study, which found that while overall cancer rates were higher in the HIV+ve population, the specific incidence of BC did not markedly differ from the HIV-ve population [22]. This finding is significant as it suggests that effective HIV management may play a role in reducing the additional cancer risk typically associated with HIV.

The influence of demographic factors such as age, race, and socioeconomic status on the incidence of BC among PLWH is profound. Younger individuals diagnosed with HIV and initiating ART present a different cancer risk profile than those diagnosed at an older age. Moreover, the racial and ethnic disparities observed in the general population continue to be evident among the HIV-infected [23]. African American women with HIV, for instance, are more likely to be diagnosed with BC at a later stage than their Caucasian counterparts [24]. This disparity is reflective of the broader issues of racial inequality in healthcare access and outcomes.

Socioeconomic status also significantly impacts BC incidence and outcomes for PLWH. Limited access to healthcare resources, including cancer screening and early detection programs, often leads to delayed diagnosis and poorer outcomes, particularly in lower socioeconomic groups. This issue is exacerbated in resource-limited settings, where both HIV and BC represent cumbersome conditions [25].

## 3. Biological Interplay between HIV Infection and Breast Cancer

Several studies showed that the age of cancer diagnosis in the HIV/AIDS population is about 20 years younger compared to the general population, and patients with HIV often present with more advanced stages of BC without any direct correlation to viral load or CD4+ T cell counts, as stated by Caccuri et al. [26]. Among PLWH, factors such as immunosuppression, chronic inflammation, altered gene expression, and prolonged exposure to ART heighten the risk of cancer development (Figure 2).

### 3.1. Immune System Impairment and Cancer Risk

Immunosuppression in HIV patients significantly increases their susceptibility to various cancers, broadly categorized as ADCs such as Kaposi sarcoma, non-Hodgkin lymphoma, and invasive cervical cancer, and NADCs, which include a wider range of tumors like lung, liver, and, increasingly, breast cancer [27]. The underlying mechanism involves the HIV-induced impairment of the immune system’s ability to detect and eliminate malignant cells. Specifically, the virus targets CD4+ T cells, leading to decreased immune competency against infections and neoplasia [28]. This immunosuppressive state allows oncogenic viruses, such as human papillomavirus and Epstein-Barr virus, to persist and potentially initiate cancer. While the direct link between HIV-related immunosuppression and BC is less clear than for virally-induced cancers, emerging research suggests that immunosuppression could contribute to a modestly increased risk of BC through diminished immune surveillance and control of oncogenic processes [29]. This relationship underscores the importance of ongoing vigilance and cancer screening in PLWH. Literature evidence supporting these observations includes studies by Grulich et al. [30], which highlight the increased incidence of cancers in people with HIV/AIDS compared to the general population, and Franceschi et al. [31], who discuss the epidemiological aspects of cancer in PLWH, noting the shift towards a broader spectrum of cancers as these patients live longer due to ART. Additionally, research by Coghill et al. [16] explores the incidence of cancer among PLWH, including BC, emphasizing the role of immunosuppression.

The research by Wu et al. [32] indicates that HIV-related weakening of the immune system impairs its ability to surveil and contain tumor cells, which can lead to unchecked growth. The measurement of CD8 has been correlated with overall survival (OS), and four indicators—the lymphocyte-monocyte ratio, platelet-lymphocyte ratio, CD3, and CD8—could predict progression-free survival. These results are supported by Spano et al. [33], who identify the value of CD3 and the lymphocyte-monocyte ratio as significant predictors for progression-free survival.

In the same way, the work of Ayeni et al. on African women with BC [34] revealed that HIV could dampen the immune response to breast tumors by altering the density of tumor-infiltrating lymphocytes within the epithelium and stroma, affecting the response to systemic therapies.

### 3.2. HIV, Chronic Inflammation, and Cancer

HIV induces chronic inflammation through several mechanisms. Firstly, the virus directly infects and depletes CD4+ T cells, a critical component of the immune system, leading to immune dysregulation. This depletion triggers a compensatory immune response, including the activation of CD8+ T cells, which in turn contributes to systemic inflammation. Secondly, HIV can persist in reservoirs within the body, even in individuals on effective ART, leading to ongoing immune activation as the body continuously attempts to combat the virus. This persistent immune activation is characterized by elevated levels of pro-inflammatory cytokines and immune activation markers. Furthermore, HIV infection disrupts the gut mucosal barrier, leading to microbial translocation [35]. This process allows bacterial products (lipopolysaccharides, LPS) to enter the bloodstream from the gut, further driving systemic inflammation. The combination of direct viral effects, immune system activation, and microbial translocation creates a state of chronic inflammation, immune activation, and immunosenescence that not only contributes to the progression of HIV disease but also to comorbid conditions, including an increased risk of cancer development [36].

Mechanistically, chronic inflammation can lead to DNA damage, promote cellular proliferation, and inhibit apoptosis, thereby increasing cancer risk. Several biomarkers of inflammation, such as C-reactive protein (CRP), TNF-alpha, and IL-6, have been associated with an increased risk of BC in the general population [37].

As regards HIV per se, the viral protein R (Vpr) and the negative factor (Nef) have been implicated in modulating cellular pathways that influence cancer cell proliferation and survival, shedding light on the indirect mechanisms through which HIV may contribute to tumorigenesis. Vpr, a multifunctional protein, is known to interfere with cell cycle regulation, inducing cell cycle arrest at the G2/M phase, which can lead to an environment conducive to cancer development. For instance, Vpr has been shown to interact with the damage-specific DNA binding protein 1 (DDB1), a component of an E3 ubiquitin ligase complex, thereby hijacking the cell’s ubiquitin-proteasome system and influencing pathways critical for DNA repair and cell proliferation [38]. On the other hand, Nef, a protein essential for viral replication and pathogenicity, plays a pivotal role in altering T cell functions and enhancing viral infectivity. Moreover, Nef’s ability to downregulate MHC-I molecules and modulate signaling pathways can lead to immune evasion and the creation of a pro-tumorigenic microenvironment [39].

### 3.3. Gene Expression Alteration and Breast Cancer in HIV Infection

The interplay between HIV infection and alterations in the expression of genes involved in cell cycle regulation, apoptosis, and DNA repair mechanisms is complex and has significant implications for cancer development, including breast cancer.

HIV infection can lead to the dysregulation of cell cycle proteins, such as cyclins and cyclin-dependent kinases (CDKs), which are crucial for the orderly progression of cells through the cell cycle, as demonstrated by Mavigner et al., who demonstrated that HIV impacts cell cycle regulation by affecting cyclin D1 expression, promoting uncontrolled cell proliferation [40].

HIV can interfere with apoptosis by modulating the expression of Bcl-2 family proteins, which are key regulators of programmed death. A study by Cummins and Badley elucidated how HIV proteins, such as Tat and Vpr, can inhibit apoptosis by altering the expression of Bcl-2 and related proteins, thereby contributing to the survival of cells with DNA damage [41].

Furthermore, HIV infection has been associated with impaired DNA repair capacity, which can lead to the accumulation of mutations and genomic instability. Deeks et al. discussed how HIV-induced inflammation leads to oxidative stress, resulting in DNA damage and reducing the efficacy of DNA repair mechanisms. This environment facilitates mutations and chromosomal aberrations in genes critical for tumor suppression and cell cycle control [42].

In addition, HIV’s upregulation of pro-inflammatory cytokines has been linked to cancer progression due to their role in promoting cell proliferation, angiogenesis, and metastasis. Several studies [43] highlight the role of chronic immune activation and inflammation in altering gene expression, which can contribute to the carcinogenesis process in PLWH [44]. Additionally, studies such as those by Clifford and Franceschi [45] have explored the molecular epidemiology of cancer in HIV patients, noting that the immunosuppression and chronic inflammation associated with HIV are critical factors in the altered gene expression patterns leading to cancer, including breast cancer. These alterations not only facilitate the initiation and progression of malignancies but also suggest potential targets for therapeutic intervention and highlight the importance of monitoring gene expression changes in HIV-infected patients as a part of their cancer risk assessment and management strategies.

While Brandão et al. [46] suggest that, maybe due to potentially lower estrogen levels, HIV+ve patients are diagnosed at a more advanced stage of BC and are less likely to have the hormone receptor positive/human epidermal growth factor 2 negative (HR+ve/HER2-ve) subtype compared to HIV-ve patients, with more challenging clinical and therapeutic management, research by Caccuri et al. [26] remarks the role of HIV matrix protein p17 in BC by enhancing clonogenic activity, angiogenesis, and lymphangiogenesis, thereby facilitating tumor growth and metastasis, by activation of the MAPK pathway through interaction with the CXCR2 receptor.

### 3.4. Debating Considerations

The scientific literature provides limited and disputing evidence regarding the hypothesis that HIV infection could have a protective role against the development of BC. A few epidemiological studies have observed lower-than-expected rates of BC among PLWH compared to the general population, suggesting a possible protective effect of HIV infection itself or its treatment.

For instance, a retrospective cohort study by Hessol et al. [47] found a lower incidence of BC among PLWH compared to national cancer incidence rates. One proposed mechanism for this observation includes the altered immune response in HIV-infected individuals, which may somehow contribute to a reduced ability of breast cancer cells to establish or grow. Additionally, the effects of ART on hormonal regulation and on the immune system have been speculated to play a role in modulating cancer risk; on the other hand, drugs like efavirenz have been shown to bind to and activate estrogen receptors in breast tissue, offering a possible mechanism for efavirenz-induced gynecomastia in HIV/AIDS patients and potentially influencing BC development. On the same pathway, studies by Goedert et al. [48] and Ruiz et al. [49] suggest that immunosuppression might play a protective role against BC, with a lower BC incidence observed in patients with AIDS, which might reflect the impact of HIV on mammary cell proliferation.

Another study by Hessol et al. [50] found that individuals infected with CXCR4-tropic HIV had a 90% lower risk of developing BC compared to those with CCR5-tropic HIV. This significant reduction in risk could explain the lower incidence of BC observed among patients with AIDS in the United States. The association suggests that HIV variants that target the CXCR4 receptor, which is more prevalent in BC cells than in normal cells, might have a protective effect against the development of BC. This study considers the possibility that CXCR4-tropic HIV could inhibit tumor-promoting macrophages and the potential for CXCR4-tropic HIV to induce apoptosis in breast cells independently of CD4 interaction, suggesting a unique mechanism by which these HIV variants might reduce BC risk. The increase in ART use and its effectiveness could influence the prevalence of CXCR4-tropic HIV, potentially contributing to the trends observed in BC incidence among PLWH.

However, it is crucial to note that these findings are controversial and not universally accepted, with other studies failing to replicate these results or suggesting that the observed lower incidence rates may be due to underdiagnosis or reporting biases in the HIV+ve population. The consensus among researchers is that more rigorous, controlled studies are needed to clarify the relationship between HIV infection and BC risk, considering the potential for confounding factors and the complex interplay of HIV with the immune system [33,50,51].

Research into the genes and molecular mechanisms associated with both HIV infection and BC suggests a potential link between the two. However, the nature of this relationship—whether protective, detrimental, or neutral regarding BC risk—is the subject of ongoing debate. Microarray analyses, such as those cited by Grover et al. [52], have identified 17 shared genes between HIV infection and BC, with ten being overexpressed and seven underexpressed in both conditions. Additional research has pointed to overlapping genes, signaling pathways, proteins, and receptors involved in both HIV and BC. While the shared genetic and receptor profiles suggest ways HIV might influence BC’s progression, they do not prove causation.

Eventually, the theory of a “breast cancer virus” [53], supported by Rakowicz-Szulczynska’s old data [54] showing 90% homology between BC DNA sequences and the HIV-1 gp41 gene, as well as Liu’s discovery [55] of a whole proviral structure incorporated into the genome of two BC, is inconclusive and lacks a solid foundation, needing more careful studies.

## 4. Breast Cancer Clinical Characteristics in PLWH

As the survival of PLWH increased, so did the probability of developing NADCs, including BC [4,56]. Literature describing the characteristics of HIV+ve BC patients has progressively accumulated over the last decades, with several reports from different parts of the world [46].

Overall, HIV+ve individuals tend to develop BC earlier in life compared to their HIV-ve counterparts. A younger age at diagnosis was consistently observed in PLWH from sub-Saharan Africa [25,34,57,58,59,60,61]. Most of these cohorts report a median age at diagnosis below 50 years, 5 to 10 years younger than the median age at diagnosis observed in HIV-ve patients [34,62,63]. Reports from the United States (US) are conflicting in terms of age at presentation. A registry-based study showed that, between 1996 and 2007, age at BC diagnosis did not differ between PLWH and the general population [17]. More recently, data from the National Cancer Data Base gathered between 2004 and 2014 pointed out that PLWH tend to present different tumor types, including BC, earlier in life [64].

Compared to HIV-ve subjects, HIV+ve patients are more frequently diagnosed with locally advanced or metastatic BC. According to a systematic review and meta-analysis of literature published before 2020, PLWH had higher odds of being diagnosed with stage III–IV BC, either if they were from sub-Saharan Africa (Odds Ratio [OR] 1.23) or North America (OR 1.76) [46]. More recently, the African Breast Cancer-Disparities in Outcomes (ABC-DO) study reported comparable rates of stage III-IV BC among 313 women living with HIV and 1184 HIV-ve women [25]. However, in the sub-Saharan population, where implementation of BC screening is scarce and healthcare resources are limited, baseline odds of having advanced BC at diagnosis are high regardless of HIV status [34,65]. On the other hand, US-based reports show that women living with HIV have a higher probability of presenting stage III/IV BC regardless of their clinic-pathological characteristics and socio-economic status [17,64].

Several studies investigated the histopathological features of BC in HIV+ve patients, with inconsistent results. A large cohort from South Africa, including 156 HIV+ve and 614 HIV-ve BC, did not show any significant difference in terms of HR and HER2 expression [57]. Similar results emerged from other, more recent cohorts, as reported by Phakathi et al. and by Chasimpa et al. [25,58]. However, other evidence claims a higher incidence of HR-ve BC in women living with HIV. In the meta-analysis from Brandāo et al., among sub-Saharan women, the OR of having a HR+ve/HER2-ve BC was 0.81 compared to HIV-ve subjects [46]. In a subsequent report, the same author reported a significant correlation between HR-ve/HER2-ve BC and HIV+ve patients compared to HIV-ve subjects from Mozambique [66]. This association could be likely due to the younger age at diagnosis of HIV+ve BC patients, since younger women have a higher incidence of HR-ve BC [67].

Data are also emerging about the molecular characterization of BC occurring in PLWH. Phakathi et al. analyzed 176 BC samples of HIV+ve patients and 200 samples of HIV-ve patients from the South African Breast Cancer and HIV Outcomes (SABCHO) cohort, using the PAM50 gene expression assay to determine molecular subtype and risk of recurrence (ROR) score. Specimens from the two groups were age-matched. No significant differences in molecular subtypes emerged between HIV+ve and HIV-ve patients, except for a greater proportion of luminal-B tumors among HIV-ve women. The risk of recurrence was comparable across the two populations, with a median score of 67, 65.5, and 68 in the whole cohort in HIV+ve and HIV-ve patients, respectively [63]. Caro-Vegaset et al. performed whole-exome sequencing of 13 BC samples diagnosed in HIV+ve patients. Compared to a cohort of 716 HIV-ve BC patients extrapolated from the Genomic Data Commons Data Portal, no difference emerged in terms of gene alterations. However, BC samples from HIV+ve patients presented a significantly higher tumor mutational burden (mean 82.6 mutations/megabase versus 4.38 mutations/megabase in the control group) [68]. Despite being preliminary, these results may have important implications for the management of BC patients with HIV and deserve further exploration in future research.

## 5. Clinical Outcomes of HIV+ve Breast Cancer Patients

Despite the fact that HIV infection does not increase the risk of developing BC, it can have a significant impact on BC outcomes [16,69]. A consolidated body of evidence, including a meta-analysis, indicates that people living with HIV and BC respond less to anticancer treatment and display worse survival outcomes compared to HIV-ve patients, especially in the non-metastatic setting, while results in the metastatic phase are conflicting [46].

In the non-metastatic setting, data from the SABCHO study suggest that women from South Africa living with HIV with stage I-III BC have lower rates of survival at 2 years compared to HIV-ve patients from the same cohort. Indeed, at a median follow-up of 29 months, 2-year OS was 72.4% among 499 HIV+ve patients and 80.1% among 1868 HIV-ve patients (hazard ratio, HR 1.49; *p* = 0.001). Reduced survival was observed in HIV+ve individuals across all the significant subgroups identified and was independent from viral load and CD4+ lymphocyte count [34]. Survival outcomes from the ABC-DO study, including 258 HIV+ve and 974 HIV-ve stage I-III sub-Saharan BC patients, were consistent with those of the SABCHO cohort. In this population, overall, 3-year-olds had higher rates of PLWH (HR 1.64), especially in patients alive beyond the 18th month of observation [25]. In a registry-based study from the US, more than 1,000,000 stage I-III BC patients were included, of which 1057 were HIV+ve. In this population, all-cause mortality was higher in HIV-infected subjects (41.7% versus 15.8%, HR 1.98) [64]. Lastly, Phakati et al. reported the survival outcomes of a cohort of 1019 BC patients from South Africa, including 221 HIV+ve patients. Among patients with stage I-III BC (*n* = 1817, 1179 HIV+ve, and 638 HIV-ve), disease-free survival (DFS) was significantly shorter in those with HIV, especially in the small subset of patients (*n* = 45) not treated with ART. In the whole cohort, including stage IV patients, HIV+ve patients not receiving ART displayed the worst OS [62].

Human immunodeficiency virus infection also seems to negatively affect the response to neoadjuvant chemotherapy (NACT). Nietz et al. retrospectively analyzed the pathological complete response (pCR) rates of patients from the SABCHO with stage I-III BC who underwent NACT and subsequent surgery [70]. Moreover, 715 women—173 HIV+ve and 542 HIV-ve—were included. Clinic-pathological characteristics were balanced in the two groups. PLWH had a significantly lower rate of pCR compared to HIV-ve subjects (8.7% versus 16.4%; OR 0.48; *p* = 0.01). This finding was consistent across all subgroups, with a marked impact of HIV+ve status on the lack of pCR in HR+ve patients compared to HR-ve patients. Additionally, pCR was not related to the use of ART, viral load, or CD4+ count [70]. Similar results emerge from a report by Martei et al., analyzing the pCR rate in a cohort of BC patients from Botswana treated with NACT (*n* = 110, of which 26 HIV+ve and 84 HIV-ve) [71]. Groups were unbalanced since more HR+ve/HER2-ve patients were HIV+ve. Only 1/20 (5%) HIV+ve BC patients with adequate pathology data achieved pCR, compared to 14/52 (21%) HIV-ve patients (OR 0.20; *p* = 0.048). Additionally, HIV+ve patients received a significantly lower relative dose intensity of chemotherapy than HIV-ve subjects (0.70 versus 0.80; *p* = 0.028) and had a numerically higher rate of early treatment discontinuation (35% versus 23%; *p* = 0.20). After a median follow-up of 20 months, the 2-year OS rate among stage III BC patients was significantly shorter in the HIV+ve group (58% versus 74%) [71].

Fewer reports focused on a population of stage IV-only BC patients. Pumpalova et al. looked at the OS of a cohort of 550 metastatic BC women from South Africa, 147 of whom lived with BC. According to their results, HIV+ve HR-ve BC patients had shorter survival compared to HIV uninfected patients (1-year OS 27.1% versus 48.8%; HR 1.94; *p* = 0.002). However, OS did not significantly differ between the two groups in the overall population or in the HR+ve population [60]. Moreover, 210 stage IV breast cancer (BC) patients were included in the ABC-DO cohort: 39 were HIV-positive (HIV+ve) and 171 were HIV-negative (HIV-ve). In this specific subpopulation, the crude 3-year survival rates for HIV-positive and HIV-negative individuals were 16.5% and 12.7%, respectively, with a Hazard Ratio (HR) of 1.10 [25]. The interaction between drugs used for BC treatment and those for HIV management complicates the therapeutic approach, potentially contributing to the observed higher mortality rates in BC patients with HIV. This phenomenon is similar to interactions observed with other classes of drugs [72]. In this contest, a lack of guidelines and education between healthcare providers often results in the substandard cancer care offered to HIV+ve BC patients [73,74]. Although extensively studied, survival differences between HIV+ve and HIV-ve BC patients need to be further investigated to elucidate the underlying biological, epidemiological, and clinical determinants and eventually improve the management of this population.

## 6. Antiretroviral Therapy and Anticancer Agents: Cross-Toxicities and Drug–Drug Interactions

Cancer patients with HIV face significant challenges as they are frequently excluded from clinical trials, resulting in a dearth of data on the potential toxicity and outcomes involving both ART and anticancer agents. Despite the heightened risks associated with cross-toxicity and drug–drug interactions (DDIs), discontinuing ART during anticancer treatment is deemed unfavorable, as it has been related to poorer outcomes [75,76]. In this context, a comprehensive assessment of the pharmacological interactions between antiretroviral regimens and anticancer agents becomes imperative. The risk of increased toxicity resulting from the simultaneous use of these classes of drugs depends on pharmacokinetic and pharmacodynamic interactions. Some comprehensive, up-to-date, evidence-based online tools are currently available to evaluate DDIs and cross-reactions between antiretroviral drugs and both anticancer cytotoxic (Figure 3) and biological agents (Figure 4) [77,78,79]. Pharmacokinetic interactions are due to drug-metabolizing enzymes [80]. It is noteworthy that the majority of antiretroviral drugs act as substrates for the cytochrome P (CYP) 450 system, functioning as either inducers or inhibitors. Additionally, other isoenzymes, including CYP1A2, CYP2C9, and CYP2D6, are commonly associated with antiviral metabolism. Given that several antiretrovirals undergo metabolism via these same enzymatic systems, co-administration with anticancer agents may not only result in drug accumulation and potential toxicity but also lead to diminished efficacy in either or both drug classes [81,82,83]. Pharmacodynamic interactions depend on the potential overlap of side effects resulting from the use of two or more molecules. Among the drugs employed against BC, cytotoxic agents are notoriously associated with neutropenia, which is also observed with zidovudine. Platinating agents, taxanes, and vinca-alkaloids are associated with peripheral neuropathy, which is also a common side effect of nucleoside reverse transcriptase inhibitors (NRTIs). Molecularly targeted agents typically exhibit a distinct toxicity profile compared to the classic side effects associated with chemotherapy, but they are not devoid of toxicities, which include cardiac toxicity, rash, hepatotoxicity, or hypertension. QT prolongation, for example, commonly observed with the protease inhibitors (PIs) atazanavir, ritonavir-boosted lopinavir, and saquinavir, is also increasingly common with the newer molecularly targeted anticancer agents, including the tyrosine kinase inhibitors [82].

Finally, interpatient variability, influenced by factors such as gender, age, genetics, and comorbid conditions, especially those affecting renal or hepatic excretion, emerges as a critical element affecting both treatment response and the toxicity profile of ART and anticancer agents [84].

Entry, Attachment, and Capsid Inhibitors

Fusion inhibitors, exemplified by albuvirtide and enfuvirtide, stand out as they are not metabolized by the CYP450 system. Consequently, their use is unlikely to lead to significant interactions with anticancer drugs [82]. Fostemsavir, a phosphonooxymethyl prodrug of temsavir and a novel HIV-1 attachment inhibitor, binds to and inhibits gp120 activity without affecting the CYP system. Though its most common toxicities include gastrointestinal symptoms, fatigue, headaches, and respiratory tract infections [85], caution is advised when combining it with certain antiblastic agents due to limited evidence [78,86].

The monoclonal antibody ibalizumab-uiyk, an innovative post-attack HIV-1 inhibitor targeting CD4, lists nausea, fatigue, pyrexia, rash, and dizziness among its main side effects [87]. Although specific studies on DDIs are lacking, their novel mechanism of action suggests minimally expected interactions [88]. Lenacapavir, a capsid inhibitor, moderately inhibits CYP3A. Due to limited data, cautious use is recommended when combining it with the majority of antineoplastic agents [78,86].

Maraviroc, a chemokine receptor 5 (CCR5) antagonist, is a substrate for CYP3A and ABCB1 but does not interfere with metabolism or transport. Because of their cross-toxicity, caution is warranted in conjunction with taxanes, tamoxifen, and ribociclib due to potential adverse events, including hepatotoxicity and intestinal symptoms [78,86].

Integrase inhibitors

Integrase inhibitors exhibit hepatic elimination and minimal interference with CYP450 enzymes. Their primary adverse effects encompass hepatitis and myalgia [80,89]. Considering that taxanes are also known for their potential to induce hepatotoxicity, co-administration with integrase inhibitors should be approached cautiously [90]. As regards cabotegravir, its long-acting nature means that potential interactions have a prolonged impact, necessitating careful consideration and monitoring [91].

Non-nucleoside reverse transcriptase inhibitors (NNRTIs)

Non-nucleoside reverse transcriptase inhibitors undergo extensive metabolism via the CYP450 enzyme and are linked to various adverse events, including rash, neurological symptoms, and an increase in transaminases. Given this shared cytochrome system metabolism and the potential for cross-toxicity, the administration of efavirenz, etravirine, and nevirapine should be approached cautiously, particularly when used concomitantly with major anticancer agents [92].

Nucleoside reverse transcriptase inhibitors (NRTIs)

Nucleoside reverse transcriptase inhibitors undergo metabolism independently of the CYP450 system and do not exhibit induction or inhibition of CYP450 enzymes. With a generally well-tolerated profile, NRTIs are not prone to DDIs via the cytochrome P450 system. However, they may be susceptible to transporter-mediated interactions, particularly through the renal pathway, which serves as their primary elimination route. Despite the overall low potential for DDIs, it is important to note that NRTIs can contribute to renal dysfunction, particularly when co-administered with nephrotoxic drugs such as cisplatin. Therefore, close monitoring of renal function is recommended during concurrent use. In instances of renal impairment, adjustments to dosage or consideration of substitution with carboplatin may be necessary to optimize patient safety and treatment efficacy [80].

Protease inhibitors (Pis)

Protease inhibitors demonstrate potent inhibitory activity on CYP450 enzymes, with ritonavir being a major inhibitor of CYP3A4 and actively inhibiting CYP2C8, CYP2D6, and ABCB1. Additionally, it weakly induces CYP2B6, CYP2C9, CYP3A4, and ABCB1 [80]. Regimens involving PIs are associated with a spectrum of adverse events, including gastrointestinal symptoms, dyslipidemia, insulin resistance, transaminase elevation, and an increased risk of cardiovascular events [92]. Full-dose ritonavir is linked to more frequent and severe hepatotoxicity compared to other PIs, but the risk is reduced with lower ritonavir doses in dual-PI combinations. Certain pIs, such as atazanavir, saquinavir, and ritonavir-boosted lopinavir, are particularly associated with QT interval prolongation. Due to their interference with the enzymatic activity of various cytochrome P450 complexes and a high toxicity profile, caution should be exercised, and PIs should be either avoided or used with care in patients receiving chemotherapy and targeted therapy for cancer.

## 7. Conclusions

The convergence of HIV infection and BC poses a significant clinical challenge, marked by the intricate dynamics of immunosuppression, chronic inflammation, altered gene expression, and the effects of ART on cancer management and risk. Our thorough review sheds light on the nuanced relationship between HIV and BC, revealing not only a slightly elevated risk of BC in those with HIV but also the likelihood of an earlier-onset and more severe disease at diagnosis. It delves into the complex mechanisms involved, such as compromised immune function, persistent inflammation, and changes in gene expression. Crucially, our analysis points out notable research deficits, especially concerning the epidemiology of BC in PLWH, how HIV affects BC prognosis, and the intricate interactions between ART and chemotherapy. These insights highlight the critical need for focused investigations to decode the molecular basis of the association between HIV and BC, enhance early detection and diagnostic methods, and forge personalized treatment plans that address the distinct obstacles encountered by those afflicted by both conditions. Tackling these research priorities promises to refine patient care and boost outcomes for this at-risk group, ultimately aiding in the broader objectives of cancer containment and HIV care management. Specifically, dedicated research is essential to unraveling the molecular connections between HIV and BC, offering valuable revelations on how HIV impacts cancer susceptibility, progression, and therapeutic outcomes. Clinically, there’s a pronounced demand for cohesive treatment regimens that can navigate the drug interactions between ART and cancer therapies, coupled with a push for more proactive BC screening and diagnostic measures among HIV-positive patients. Moreover, formulating specialized treatment strategies that take into account unique challenges such as immunosuppression and chronic inflammation is imperative. Additionally, enhancing the awareness and understanding of healthcare providers regarding the care of patients with both HIV and BC is vital, as is educating patients about the significance of routine screening and the impact of their dual diagnosis on treatment options.

## Figures and Tables

**Figure 1 ijms-25-03222-f001:**
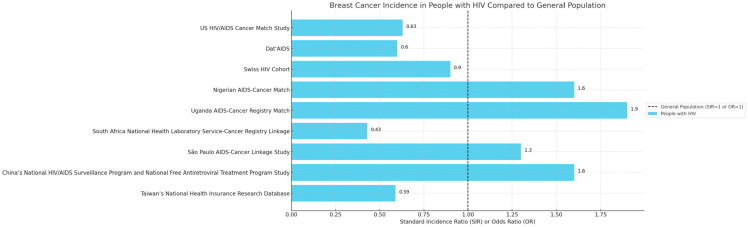
Breast cancer incidence in PLWH compared to people without HIV. Each bar represents a different cohort study’s standard incidence ratios (SIRs) or odds ratios (ORs) for breast cancer incidence in people with HIV. The dashed line at x = 1 represents the baseline risk for the general population.

**Figure 2 ijms-25-03222-f002:**
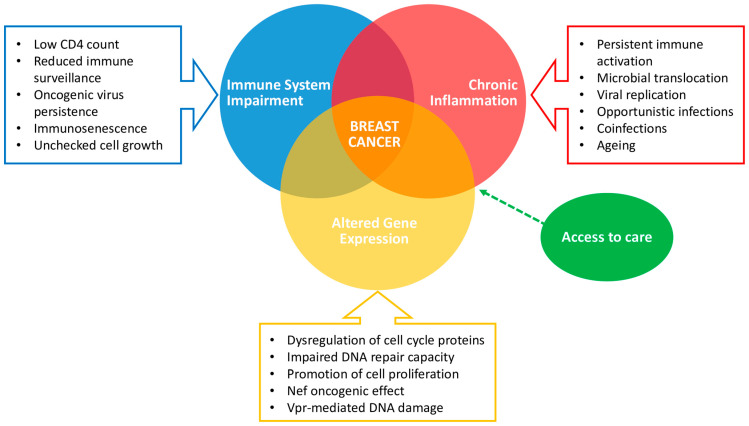
Interplay between HIV and breast cancer. Impaired immune systems, chronic inflammation, and altered gene expression play a key role in breast cancer development in PLWH, whereas access to care could make a difference in diagnosis and clinical management.

**Figure 3 ijms-25-03222-f003:**
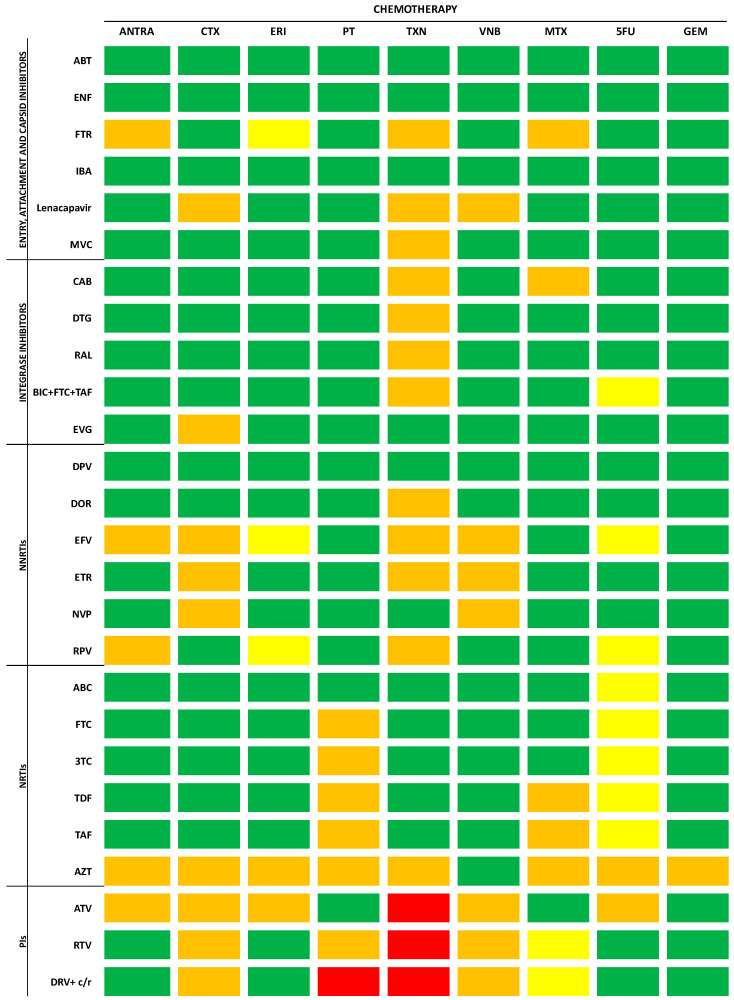
Drug–drug interactions between ART and cytotoxic agents according to Medscape, Liverpool University, and PEPID interaction checkers’ reports. Legend: no interaction expected (green); potential weak interaction (yellow); potential interaction (orange); do not co-administer (red). ANTRA: anthracyclines; CTX: cyclophosphamide; ERI: eribulin; GEM: gemcitabine; MTX: methotrexate; PT: platinum salts; TXN: taxanes; VNB: vinorelbine; 5FU: 5-fluorouracil; NNRTIs: non-nucleoside reverse transcriptase inhibitors; NRTIs: nucleoside reverse transcriptase inhibitors; PIs: protease inhibitors; ABC: abacavir; ABT: albuvirtide; ATV: atazanavir; AZT: zidovudine; BIC: bictegravir; CAB: cabotegravir; c: cobicistat; DPV: dapivirine; DRV: darunavir; DOR: doravirine; DTG: dolutegravir; EFV: efavirenz; ENF: enfuvirtide; EVG: elvitegravir; ETR: etravirine; FTC: emtricitabine; FTR: fostemsavir; IBA: ibalizumab-uiyk; MVC: maraviroc; NVP: nevirapine; RAL: raltegravir; RPV: rilpivirine; RTV/r: ritonavir; TAF: tenofovir alafenamide; TDF: tenofovir-disoproxil fumarate; 3TC: lamivudine.

**Figure 4 ijms-25-03222-f004:**
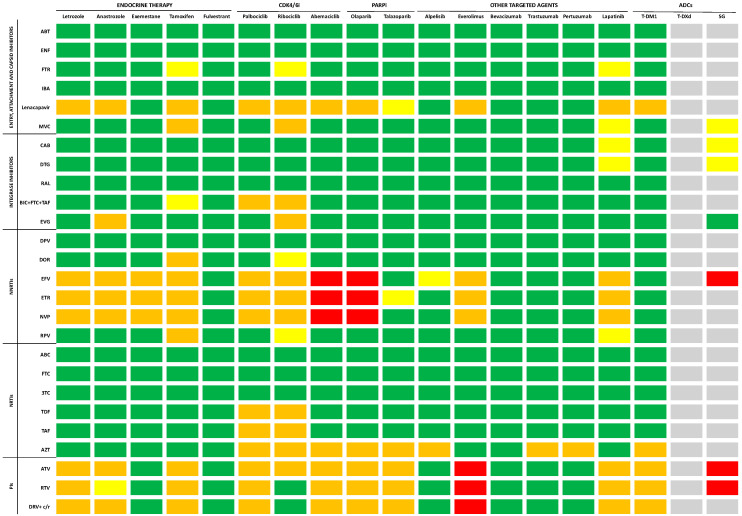
Drug–drug interactions between ART and biological anticancer agents according to Medscape, Liverpool University, and PEPID interaction checkers’ reports. Legend: no interaction expected (green); potential weak interaction (yellow); potential interaction (orange); do not co-administer (red); unknown interactions (grey). ADCs: antibody drug conjugates; CDK4/6i: cyclin-dependent kinase 4/6 inhibitors; PARPi: poly(ADP) ribose polymerase inhibitors; SG: sacituzumab govitecan; T-DM1: trastuzumab emtansine; T-Dxd: trastuzumab deruxtecan; NNRTIs: non-nucleoside reverse transcriptase inhibitors; NRTIs: nucleoside reverse transcriptase inhibitors; PIs: protease inhibitors; ABC: abacavir; ABT: albuvirtide; ATV: atazanavir; AZT: zidovudine; BIC: bictegravir; CAB: cabotegravir; c: cobicistat; DPV: dapivirine; DRV: darunavir; DOR: doravirine; DTG: dolutegravir; EFV: efavirenz; ENF: enfuvirtide; EVG: elvitegravir; ETR: etravirine; FTC: emtricitabine; FTR: fostemsavir; IBA: ibalizumab-uiyk; MVC: maraviroc; NVP: nevirapine; RAL: raltegravir; RPV: rilpivirine; RTV/r: ritonavir; TAF: tenofovir alafenamide; TDF: tenofovir-disoproxil fumarate; 3TC: lamivudine.

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
