# Peer review of "Navigating the Nexus: HIV and Breast Cancer—A Critical Review"

_ijms, 2024, doi:10.3390/ijms25063222_

Round 1

Reviewer 1 Report

Comments and Suggestions for Authors

Minor revision is needed.

Comments on the Quality of English Language

No issue with language presentation.

Author Response

Major Comments

  1. Include the prevalence, incidence and mortality of breast cancer around the world. Reply: We added what you suggested
  1. Highlight the percentage of breast cancer observed in HIV infected individuals. Reply: Done
  1. Please include future implications as well. Reply: We rephrased the conclusions section adding what you suggested
  1. Please include a table highlighting the incidence of NADCs in people with HIV+ and HIV- by geography. [refer PMID: 34087151]. Reply: We added the reference you suggested. In addition, we added a graph about breast cancer incidence in HIV+ and HIV-. We decided to add only data about breast cancer since this is the subject of our review.
  1. It would be good to provide a systematic analysis of cohort reports published based on different geographical locations. Reply: As we stated, we added a graph about that, better highlighting breast cancer epidemiology in PLWH.
  1. Authors could include clinical trials data on ART and anticancer agents. Reply: We have attempted to cover every aspect of ART and DDIs. Furthermore, following another reviewer's suggestion, we have included a statement about long-acting therapies. We have been cautious not to overload the review with information about ART, as it constitutes only one aspect of our analysis. We hope this meets your expectations.
  1. Refer articles [PMID: 31804663]. Reply: We added the references you suggested.

Minor Comments

  1. Line 150: No need to abbreviate HPV and EBV. Use abbreviations if the intended word is used for more than thrice. Reply: Done

Reviewer 2 Report

Comments and Suggestions for Authors

Please see the attached file for review comments.

Author Response

I suggest to mention a novel, long-acting antiretroviral therapy which will become a standard treatment option for people living with HIV (please see Nachega, J.B., et al. Long-acting antiretrovirals and HIV treatment adherence. Lancet HIV, 2023. 10: e332–342).

Reply: We added few lines about cabotegravir in the INSTI paragraph, citing the ref you suggested.

Lines 75-84: No relevant references are cited in this paragraph.

Reply: We added references

Lines 85-87: Please verify the correctness of the information based on the cited article by Zucchetto et al. It is hard to find breast cancer data in this article.

Reply: We changed the text you highlighted, fixing the reference problem

Line 89: A necessary space is missing. It should be “[13] identified” instead of “[13]identified”.

Reply: Done

Line 93: “Shiels et al. [14]” is an incorrect citation. Reference [14] is the article by Sung et al., not by Shiels et al. The article by Shiels et al. is listed as reference no. 56. Moreover, no data on HIV are reported in Ref. [14}. This very important issue should be addressed.

Reply: We fixed what you pointed out

Line 123: A necessary punctuation mark is missing. It should be “[14]. This” instead of “[14] This”.

Reply: Done

Line 163: The author’s name should be corrected in the citation. It should be “Wu et al. [26]” instead of “Qian Wu et al. [26]”.

Reply: Done

Line 185: It should be “lipopolysaccharides” instead of “Lipopolysaccharides”.

Reply: Fixed

Line 241: Please compare the spelling of the author’s name with that listed in the References section.

Reply: Fixed

Lines 290-291: Please unify the spelling of “overexpressed” and “under expressed” (together or separately).

Reply: Done

Lines 297-299: The relevant reference should be provided for the statement “as well as Liu's discovery of a whole proviral structure incorporated into the genome of two BC, is inconclusive and lacks a solid foundation, needing more careful studies”. The article by Liu et al. should be included in the References section.

Reply: Done

Line 358: It should be “compared to HIV-ve patients” instead of “compared do HIV-ve patients”.

Reply: Fixed

Line 364: No italics should be used. I suggest “hazard ratio” instead of “hazard ratio”.

Reply: Done

Line 372: It should be “et al.” instead of “et coll.”.

Reply: Done, in that line and in the others where it was reported

Lines 374-375: The statement “Among patients with stage I-III BC (n=817, 1179 HIV+ve and 638 HIV-ve)” should be modified (1179+638=1817). The number of total BC or HIV+ve patients is incorrect. This important issue should be addressed.

Reply: Fixed

Line 377: I suggest “patients not receiving ART displayed” instead of “patients not received ART displayed”.

Reply: Fixed

Reviewer 3 Report

Comments and Suggestions for Authors

The authors reviewed the current evidence on the epidemiology of breast cancer among HIV-infected individuals. Overall the manuscript is well-written and this is a comprehensive review. I have a few comments that I think could help strengthen the presentation.

  • In lines 85-97, please include the country and the race/ethnicity of the study population when describing each study.
  • Please provide literature evidence for statement in lines 118-119.
  • Please provide literature evidence for lines 125-129.
  • For figure 1, please add labels to show what the different colors of squares mean.
  • Lines 176-207 lack evidence. Please include references for each of the statement in these paragraphs.

Author Response

In lines 85-97, please include the country and the race/ethnicity of the study population when describing each study.

Reply: We changed the text you highlighted

Please provide literature evidence for statement in lines 118-119.

Reply: Done

Please provide literature evidence for lines 125-129.

Reply: Done

For figure 1, please add labels to show what the different colors of squares mean.

Reply: There is no specific meaning associated with the colors of the squares; they are simply a choice of design

Lines 176-207 lack evidence. Please include references for each of the statement in these paragraphs.

Reply: Done

Round 2

Reviewer 2 Report

Comments and Suggestions for Authors

In the cover letter, the Authors of the article “Navigating the Nexus: HIV and Breast Cancer - A Critical Review“ have not responded to General Comments no. 1-4 and Specific comments no. 24-55 of the Review report (no appropriate corrections have been made in the revised version of the manuscript).

For details, please see the attached file of the review report containing these reviewer’s comments highlighted in yellow. For convenience, the appropriate line numbers in the revised version of the manuscript have been indicated by the reviewer.

Reviewer 3 Report

Comments and Suggestions for Authors

- Please provide references for lines83-88, lines 192-198. 

- Please include what years the study data were derived from in lines 93-96.

- Please reformat or remove Figure 2. Figure 2 is a bit confusing. You listed the risk factors for immune system impairment and breast cancer. I don't see how this shows the interplay between immune system impairment and breast cancer. 

Author Response

Please provide references for lines83-88, lines 192-198. 

Reply: Done

Please include what years the study data were derived from in lines 93-96.

Reply: Done

Please reformat or remove Figure 2. Figure 2 is a bit confusing. You listed the risk factors for immune system impairment and breast cancer. I don't see how this shows the interplay between immune system impairment and breast cancer. 

Reply: Dear Reviewer, thank you for your feedback regarding Figure 2. We apologize for the quality issue, which resulted from a conversion error on the MDPI platform. In our original submission, the figure was of high quality. We worked to resolve this graphic issue. Thank you once again for bringing this to our attention.

Round 3

Reviewer 2 Report

Comments and Suggestions for Authors

Please see the attached file for review comments.

Author Response

Dear reviewer, we apologize for having overlooked some points in the revision process. We have made the missing changes:

35. We have further implemented the font size in both figures. In particular, in Figure 3, this required abbreviating the names of the chemotherapic agents into their respective abbreviations, which have been appropriately explained in the caption.

45. Done.

51. Corrected.

55. Corrected.

Again, thank you for your valuable contribution to the drafting of the paper.

Reviewer 3 Report

Comments and Suggestions for Authors

The authors have adequately addressed my comments. Therefore, I have no further comments.  

Author Response

Dear reviewer, thank you for your valuable contribution to the drafting of the paper.